# LEARNING TASK RELATIONS FOR TEST-TIME TRAINING

## ABSTRACT

Generalizing deep neural networks to unseen target domains presents a major challenge in real-world deployments. Test-time training (TTT) addresses this issue by using an auxiliary self-supervised task to reduce the gap between source and target domains caused by distribution shifts. Previous research relies on the assumption that the adopted auxiliary task would be beneficial to the target task we want to adapt. However, this situation is not guaranteed as each task has a different objective, thus adaptation relies on the relation between the tasks. This limitation has motivated us to introduce a more generalized framework: Task Relation Learning for Test-time Training (TR-TTT), which can be applied to multiple tasks concurrently. Our key assumption is that task relations are crucial information for successful test-time training, and we capture these relations using a Task Relation Learner (TRL). We model task relations as conditional probabilities by predicting the label of a target task based on the latent spaces of other task-specific features. By leveraging these relations, the network can more effectively handle distribution shifts and improve post-adaptation performance across various tasks—both classification and regression—unlike previous methods focused mainly on simple classification. To validate our approach, we apply TR-TTT to conventional multi-task benchmarks, integrating it with the traditional TTT experimental protocol. Our empirical results demonstrate that TR-TTT significantly outperforms state-of-the-art methods across a range of benchmarks.

## 1 INTRODUCTION

Assuming the data distributions are identical between training and test-time, deep learning networks demonstrate robust performance across a range of tasks. Unfortunately, real-world scenarios rarely allow for such assumptions, making it challenging to apply numerous deep learning methods in practice. Solving the distributional gap between training and test-time has been emerged as a new challenge towards developing deep learning methodologies, motivating the development of domain adaptation or domain generalization. These methodologies, however, adapt or generalize to the fixed target distribution, which leads to the same challenges in real-world scenarios that the aforementioned settings face. To address these challenges, test-time adaptation (TTA) and test-time training (TTT) have emerged as the latest approaches to suppress the performance degradation caused by distributional gaps, aligning the pre-trained networks with target domains during test-time.

Both TTA (Wang et al., 2020; Nguyen et al., 2023) and TTT (Sun et al., 2020; Liu et al., 2021; Gandelsman et al., 2022; He et al., 2022; Mirza et al., 2023; Osowiechi et al., 2023; 2024) have access to a subset of the target domain during test-time, which allows them to update the network to better adapt to the target domain. In particular, TTT methods include an additional branch for auxiliary tasks, which leverage the information trained on source domain to the adaptation on the target distributions. By selecting self-supervised or unsupervised schemes for these auxiliary tasks, TTT methods effectively improve their ability to handle distributional shifts during test time.

However, it is almost impossible to pre-emptively decide which information would be effective in reducing the domain gap, as we do not have access to the ground truth of the target distribution. Thus, most existing TTT approaches rely on assumptions about which information will be beneficial for narrowing the distributional gap, making the selection of the auxiliary task a critical aspect of designing a TTT method. However, the chosen auxiliary task does not guarantee performance

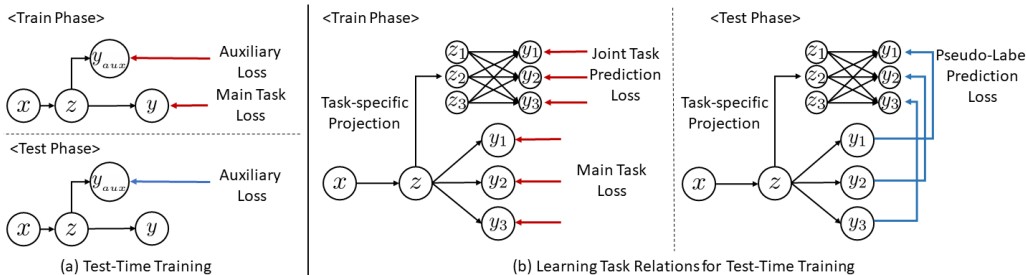

Figure 1: A comparison of conventional TTT approaches and the proposed TTT by learning Task Relations (TR-TTT). The network takes the input data $x$, encodes it into a latent representation $z$, and produces the task output $y$. (a) In the training phase, conventional TTT methods use both the main task loss and the auxiliary loss with output $y_{aux}$. In the test phase, they rely solely on the auxiliary loss. (b) In contrast, the proposed TR-TTT learns task relations that are generalizable across domain changes. In both the training and test phases, it projects the latent $z$ into the task-specific latent $z_i$. Using the projected latent vectors, TR-TTT learns task relations by predicting task outputs from other tasks' latent vectors in a separate branch (Joint Task Prediction). During the training phase, both outputs are supervised with ground truth from the source domain, while in the test phase, joint task prediction are supervised with the main task output.

improvements during test-time, as its objective is fundamentally different from that of the main task. Consequently, previous TTT methods, as illustrated in Fig. 1-(a), exhibit varying adaptation abilities depending on the relations between the main task and the selected auxiliary task. Motivated by this limitation, we take a new approach to TTT, based on the assumption that the relations between tasks are the key factor, robust on domain shift, which is effectively reducing the domain gap. This eliminates reliance on auxiliary tasks, preventing the performance degradation of the main task.

In this context, we propose a new approach for Test-Time Training by learning Task Relations (TR-TTT) , which effectively reduces the domain gap across multiple tasks, as shown in Fig. 1-(b). We argue that the relations between tasks represent crucial information that can generalize across different domains. To capture these task relations, we introduce a dedicated branch separate from the main task prediction decoders. We define inter-task relations as conditional probabilities—specifically, the probability of predicting a target task label given the set of all task-specific latent vectors. These task-specific latent spaces are projected from a shared latent space and are optimized for each task. To model these relations, we introduce a Task Relation Learner (TRL), which predicts task labels using task-specific latent vectors. To further enhance the generalizability of the task relations learned by TRL, we incorporate the masking technique motivated by the Masked AutoEncoder (MAE) (He et al., 2022). We assume that the mutual information encoded by these task relations is preserved across domain shifts and that TRL is sufficiently trained to predict task labels effectively, even with masked latent vectors. At test-time, the predicted output from masked latent vectors is guided by the predicted output from unmasked latent vectors to reduce the domain gap based on the learned task relations in the source domain.

To evaluate the influence of different TTT methods on various tasks, we utilize several multi-task benchmarks that include both classification and dense regression tasks. This setup is more challenging than previous TTT research, which typically focuses on simple classification tasks. By allowing access to multiple task labels in the source domain, each TTT method can learn more generalizable features across domain gaps. Therefore, we also enable previous TTT methods to access the ground truths for multiple tasks in the source domain. We demonstrate that the proposed TR-TTT outperforms previous TTT methods by effectively improving the performance across all tasks, as evidenced by experiments conducted on TTT between diverse datasets such as NYUD-v2, Pascal-Context, and Taskonomy. The key contributions of this work are as follows:

- We propose a novel Test-Time Training method, called TR-TTT, which utilizes task relations as key information to address domain shifts during test-time adaptation.

- Under a plausible assumption that the quantity of information encoded in inter-task relations is preserved across domain shifts, we provide a theoretical explanation of how the proposed TR-TTT objective reduces task loss in the target domain.

- We demonstrate the validity of TR-TTT for both regression and classification tasks using multi-task benchmarks. Our approach captures the relationships between tasks to enhance performance, achieving state-of-the-art results on various domain shift scenarios.

## 2 RELATED WORK

**Test-Time Adaptation & Training.** Adapting deep neural networks to a target domain is challenging due to the necessity of additional burdens for collecting and labeling data in that domain. Recent research has focused on using unlabeled data to infer the target domain's distribution, thereby narrowing the gap between source and target domains during adaptation (Liang et al., 2024). Test-time adaptation (TTA) and test-time training (TTT), which enable online adaptation, show broad applicability. A pivotal contribution in this area is TENT (Wang et al., 2020), which uses entropy as an adaptation objective for image classification. In the field of computer vision, various TTA methods have been suggested to adapt off-the-shelf models during the testing phase, focusing on tasks such as image classification (Wang et al., 2022a; Iwasawa & Matsuo, 2021; Chen et al., 2023a), semantic segmentation (Zhang et al., 2022a; Volpi et al., 2022; Lee et al., 2024), and object detection (Fan et al., 2024). However, previous TTA methods primarily enable adaptation for classification tasks, limiting their applicability to a wider range of downstream tasks. On the other hand, TIPI (Nguyen et al., 2023) enforces transformation invariance, a technique commonly used in unsupervised learning to simulate domain shifts, enabling adaptation for regression tasks, although its effectiveness on these tasks has not been fully demonstrated.

TTT employs a separate self-supervised task branch (Sun et al., 2020) as an auxiliary task for the main task adaptation, drawing inspiration from multi-task learning (Caruana, 1997). Gandelsman et al. (2022) utilizes the masked autoencoder (He et al., 2022), demonstrating its generalizability in handling distribution shifts during deployment. TTT++ (Liu et al., 2021) preserves the statistical information of the source domain to align the test-time features through contrastive learning. As an auxiliary self-supervised branch, TTT-MAE (Gandelsman et al., 2022) adopt Masked Autoencoder to adapt the network in the test-time domain while the normalizing flow (Rezende & Mohamed, 2015) has been used for Osowiechi et al. (2023). ActMad (Mirza et al., 2023) directly aligns the activation statistics of the test-time domain to the training domain directly, using the L1 norm. For only the classification task, several TTT methods (Su et al., 2022; Hakim et al., 2023; Li et al., 2023) adaptively update prototype clustering for each class, aligning the distribution shift. NC-TTT (Osowiechi et al., 2024), the most recent TTT study, adapt model on new domain by learning to classify noisy views of projected feature maps.

**Task Relations.** Capturing inter-task relations has been the main approach in the multi-task learning (MTL) domain (Caruana, 1997). In particular, partially labeled MTL addresses the challenge of inferring distribution shifts from one task to another when access to task labels is limited during training. Early studies in this field (Liu et al., 2007; Zhang & Yeung, 2009; Wang et al., 2009), uses semi-supervised learning approaches to infer these task relations. Recent works (Imran & Terzopoulos, 2019; Huang et al., 2020; Latif et al., 2020) have been applied to various domains, including computer vision and speech recognition, utilizing evolving deep neural networks. Zamir et al. (2020); Lu et al. (2021); Saha et al. (2021) directly utilize task relations by leveraging the unique characteristics of each task. Lu et al. (2021) regularizes the results of depth estimation and normal vector estimation, using the fact that normal vectors can be obtained by differentiating depth information. Saha et al. (2021) infers task relations between semantic segmentation and depth estimation, inspired by the human perception process, which uses depth information to infer semantic details. On the other hand, several studies infer distribution shifts among different tasks without explicitly analyzing task relations. Chen et al. (2020) utilizes consistency loss between similar tasks in a shadow detection problem set. Wang et al. (2022b) uses intra-domain and inter-domain adversarial loss to align the learning process of the same task across different domains. Li et al. (2022) learns pairwise task relations by regularizing the outputs of tasks from different paths in a pairwise task mapping. Nishi et al. (2024) constructs the joint-task latent space by encoding and decoding the stacked labels of multiple tasks at once. (Ye & Xu, 2024) enhances the diffusion model (Ho et al., 2020) for multi-task learning by sharing the information across the different tasks.

## 3 METHODS

### 3.1 PROBLEM DEFINITION

Consider a domain defined by the joint distribution $p_\theta(x, y)$ with random variables $\{\mathcal{X}, \mathcal{Y}\}$, where the input data $x \sim \mathcal{X}$ and the corresponding label $y \sim \mathcal{Y}$. In the source domain $\{\mathcal{X}_s, \mathcal{Y}_s\}$, a deep learning network with parameters $\theta$ is trained to learn the conditional distribution $p_\theta(y_s|x_s)$, where $x_s \sim \mathcal{X}_s$ and $y_s \sim \mathcal{Y}_s$. The goal of Test-Time Training (TTT) is to find the conditional distribution $p_\theta(y_t|x_t)$ in the target domain $\{\mathcal{X}_t, \mathcal{Y}_t\}$ by adapting the network parameters $\theta$ to the target domain, where $\mathcal{X}_s \neq \mathcal{X}_t$, without direct access to the target domain labels $y_t \sim \mathcal{Y}_t$. For classification tasks, both domains share the same label space $\mathcal{Y}_s = \mathcal{Y}_t$. We assume a similar setup for regression tasks, as the accuracy of regression tasks can be evaluated using scale-invariant metrics. Therefore, we use $\mathcal{Y}$ to represent the task label for both domains. Most recent multi-task architectures (Riquelme et al., 2021; Zhang et al., 2022b; Fan et al., 2022; Mustafa et al., 2022; Chen et al., 2023b; Huang et al., 2024) use a shared encoder across different tasks, generating a common latent space. We denote this latent space as $z_s \sim \mathcal{Z}_s$ for the source domain and $z_t \sim \mathcal{Z}_t$ for the target domain, respectively.

### 3.2 CONNECTING TASK RELATIONS FROM SOURCE TO TARGET

In a multi-task setting, both the source and target domains are expanded to $\{\mathcal{X}, \{\mathcal{Y}_i\}_{i=1}^n\}$, where $n$ is the number of tasks. Similarly, the latent space that benefits each task varies across tasks. We partition the latent space $\mathcal{Z}$ into task-specific subspaces $\{\mathcal{Z}_i\}_{i=1}^n$, where each $\mathcal{Z}_i$ represents the projection of the shared latent space $\mathcal{Z}$ that is advantageous for target task $i$, resulting in $(\mathcal{Z}_1 \cup \mathcal{Z}_2 \cup \ldots \cup \mathcal{Z}_n) \subseteq \mathcal{Z}$. For simplicity, we denote the $(\mathcal{Z}_1 \cup \mathcal{Z}_2 \cup \ldots \cup \mathcal{Z}_n)$ as $\hat{\mathcal{Z}}$.

Since we cannot access the ground truth of the target domain during adaptation, it is nearly impossible to predict in advance which specific information from the source domain will be useful for the target domain. As a result, most previous TTT approaches rely on assumptions about which information would be beneficial across domains and suggest learning strategies tailored to their specific objectives. Motivated by the previous research in multi-task learning, which uses task relation to improve the generalizability of networks, we assume that the relations between tasks are key information that can be generalized across different domains. Our first assumption is that inter-task relations will remain consistent across domain shifts, as illustrated in Assumption 1.

**Assumption 1** (**Preservation of Task Relations**). *The mutual information between the latent space and the task labels is preserved across the source and target domains, such that*

$$I(\hat{\mathcal{Z}}_s, \mathcal{Y}_i) = I(\hat{\mathcal{Z}}_t, \mathcal{Y}_i)$$

*for all $i \in \{1, 2, \ldots, n\}$, where $\hat{\mathcal{Z}}_s = (\mathcal{Z}_{s,1} \cup \mathcal{Z}_{s,2} \cup \ldots \cup \mathcal{Z}_{s,n})$ and $\hat{\mathcal{Z}}_t = (\mathcal{Z}_{t,1} \cup \mathcal{Z}_{t,2} \cup \ldots \cup \mathcal{Z}_{t,n})$.*

Consider the scenario where we predict the task label $y_i$ using task-specific features $\{z_i\}_{i=1}^n$. According to Assumption 1, the information required to predict the task label from the task-specific latent space—represented by the random variable $p(y_{s,i}|z_{s,1}, z_{s,2}, \ldots, z_{s,n})$—remains consistent across domain shifts.

As the Masked Autoencoder (MAE) (He et al., 2022) has shown outstanding performance in capturing a generalizable latent space of input data distributions, we adapt it to approximate the probability $p(y_{s,i}|z_{s,1}, z_{s,2}, \ldots, z_{s,n})$ for capturing inter-task relations. To adapt MAE for our purposes, we mask the task-specific features $z_i$ with mask $\mathcal{M}_i$, where each masked task-specific feature is represented as $\tilde{z}_i \sim \tilde{\mathcal{Z}}_i$, and their union is denoted as $\hat{\tilde{\mathcal{Z}}} = (\hat{\tilde{\mathcal{Z}}}_1 \cup \hat{\tilde{\mathcal{Z}}}_2 \cup \ldots \cup \hat{\tilde{\mathcal{Z}}}_n)$. These masked features are then used to jointly predict the task labels. We refer to this as Task Relation Learner (TRL). Our second assumption is that the TRL is sufficiently trained to produce final predictions with the masked task-specific latent space.

**Assumption 2** (**Sufficient Training of the Task Relation Learner**). *If the task relation learner is sufficiently trained, it can reliably generate task labels from the masked latent space, ensuring that*

$$I(\hat{\mathcal{Z}}, \mathcal{Y}_i) = I(\hat{\tilde{\mathcal{Z}}}, \mathcal{Y}_i)$$

*for all $i \in \{1, 2, \ldots, n\}$.*

In Assumption 2, we assume that the Task Relation Learner effectively learns task relations by capturing the mutual information $I(\hat{\mathcal{Z}}, \mathcal{Y}_i)$. This is done by approximating the probability $p(y_{s,i} | \tilde{z}_{s,1}, \tilde{z}_{s,2}, \ldots, \tilde{z}_{s,n})$, which predicts task labels from the task-specific latent vectors.

To define the optimization objective for our TTT strategy, we begin by measuring the distance, using any metric $d$, between the learnable network parameters $\theta$ and the ideal probability for predicting the target task label, represented as $d(\theta, p(\{z_{t,i}\}_{i=1}^n, y_j))$. Then, we can bind it as follows:

**Proposition 1.** *Under Assumption 1 and 2, we have*

$$d(\theta, p(\{z_{t,i}\}_{i=1}^n, y_j)) \leq d(\theta, p(\{\tilde{z}_{t,i}\}_{i=1}^n, y_j)) \tag{1}$$

$$+ \mathbb{E}_{p(\{z_{t,i}\}_{i=1}^n, \{\tilde{z}_{t,i}\}_{i=1}^n)}[d[p_\theta(y_j | \{z_{t,i}\}_{i=1}^n), p_\theta(y_j | \{\tilde{z}_{t,i}\}_{i=1}^n)]] \tag{2}$$

The left-hand side of the inequality represents the loss for task $j$, which we aim to minimize. This is equivalent to the supervised learning objective on the target domain, where we maximize the information between all available latent vectors $\{z_{t,i}\}_{i=1}^n$ and the target label $y_j$. The first term on the right-hand side, eq. (1), represents the loss when using the masked latent vectors $\{\tilde{z}_{t,i}\}_{i=1}^n$. The second term, eq. (2), reflects the gap between predicting the task label with the full latent space and its masked version.

If task relations can be effectively learned from the process of predicting task labels from masked task-specific features, we can assume that task-specific feature masking serves as an efficient tool for learning generalizable task relations in latent space across domain shift. This would allow us to transform $p(\{\tilde{z}_{t,i}\}_{i=1}^n, y_j)$ back to $p(\{\tilde{z}_{s,i}\}_{i=1}^n, y_j)$. Therefore, our training objective is to minimize $d(\theta, p(\{\tilde{z}_{s,i}\}_{i=1}^n, y_j))$, which supervises the predicted task labels derived from masked latent vectors using ground truth. Consequently, we train the network to reduce the gap between predictions made from the masked latent spaces, using ground truth as guidance during training.

Our objective during test-time is given in eq. (2), and the multi-task version is as follows:

$$\min_\theta \sum_{j=1}^n \mathbb{E}_{p(\{z_{t,i}, \tilde{z}_{t,i}\}_{i=1}^n)}[d[p_\theta(y_j | \{z_{t,i}\}_{i=1}^n), p_\theta(y_j | \{\tilde{z}_{t,i}\}_{i=1}^n)]] \tag{3}$$

During test-time, we do not have access to the ground truth for each task label. Therefore, we minimize the gap between the probability of task predictions made using the set of task-specific latent vectors $\{z_{t,i}\}_{i=1}^n$ and the predictions made using the masked vectors $\{\tilde{z}_{t,i}\}_{i=1}^n$ in the target domain. The detailed derivation of proposition 1 can be found in Appendix A.

## 3.3 Test-Time Training by Learning Task Relations

Following the previous derivation, we implement a methodology for test-time training (TTT) by capturing task relations on both training and test-time, as illustrated in the Fig. 2. The proposed framework consists of two branches: the branch for the main target task and the other branch, including the TRL, which encourages the framework to learn the generalizable task relations. Since the ideal latent space for capturing task relations may differ from that for predicting outputs, we implement the separate branch to extract each task-specific latent space. The encoder, $p_\theta(z|x)$, extracts a latent vector, $z$, from the input image, $x$. The final main output, $\{y_i^{main}\}_{i=1}^n$, are derived by passing the latent vector, $z$, through a decoder, $p_\theta(y|z)$, and supervised with the ground truth, $\{y_i\}_{i=1}^n$. On the other branch, the TRL captures the task relation by predicting the task outputs from a set of task-specific latent vectors, $\{z_i\}_{i=1}^n$. The set of the latent vectors are the projected versions of the latent vector $z$ that pass through the task-specific projection layer.

**Task-specific Projection.** The additional task-specific layers are used to project the latent vectors $z$ into the corresponding task-specific vectors, $\{z_i\}_{i=1}^n$. The task-specific projection layers consist of two layers: one for projecting the latent vector into the task-specific vector and the subsequent layer for predicting the task outputs, $y_i^{TP}$. During training, this output is supervised with the ground truth to train the task-specific projection layers:

$$\mathcal{L}_s^{TP} = \sum_{i=1}^n \mathcal{L}_i(y_i^{TP}, y_i) \tag{4}$$

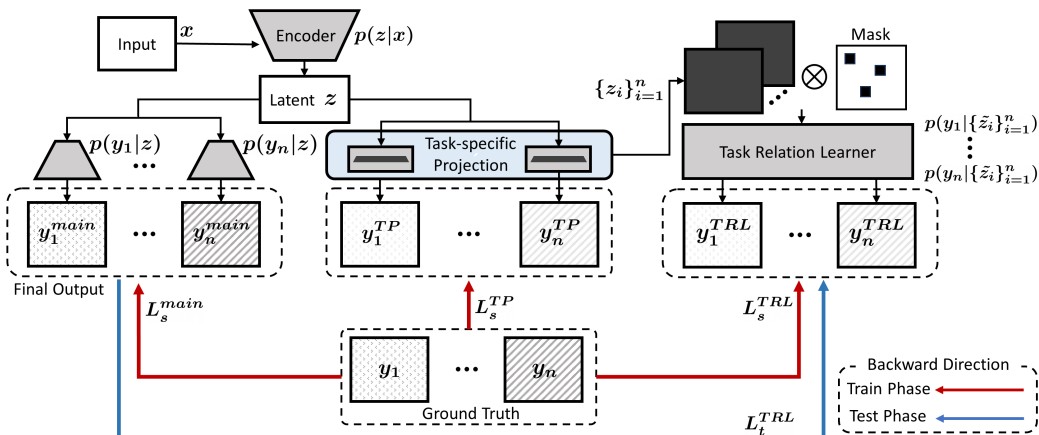

Figure 2: Overall Framework. Give input $x$, the shared encoder $p(z|x)$ encodes the latent $z$ and the task-specific decoder $p(y_i|z)$ for task $i$ decodes it into output $y_i^{main}$. Through the single-layer task-specific projection, the latent $z$ is projected into task-specific latent $z_i$, which are stacked and passed to the TRL or passed one more layer to predict another output $y_i^{TP}$. The TRL $p(y_i|\{\tilde{z}_i\}_{i=1}^n)$ predicts the output $y_i^{TRL}$ using the masked latent $\tilde{z}_i$. In the train phase, each output is supervised with the ground truth $y_i$ using the loss $\mathcal{L}_s^{main}$, $\mathcal{L}_s^{TP}$ and $\mathcal{L}_s^{TRL}$, respectively. In the test phase, only $\mathcal{L}_t^{TRL}$ is minimized to train the framework which makes $y_i^{TRL}$ close to the $y_i^{main}$.

By using task-specific projection loss, $\mathcal{L}^{TP}$, it is able to contain different task-specific information into the task-specific latent vectors. In the test-time, the trained projection layers extract the task-specific latent vectors in the target domain for the TRL.

**Task Relation Learner.** To reach the goal of TTT, the TRL, $p_\theta(y_i|\{\tilde{z}_i\}_{i=1}^n)$, is suggested to learn task relations. Similar to MAE, the TRL is implemented with vision transformer which predicts the both masked and unmasked regions of task label. The main difference is that TRL uses attention between task-specific tokens encoded from masked task-specific latent vectors, $\{\tilde{z}_i\}_{i=1}^n$. In the training phase, the outputs from the TRL, $y_i^{TRL}$, are supervised with the Joint Task Prediction Loss, $\mathcal{L}^{TRL}$, using the ground truth, $y_i$. The loss consists of the supervision loss for each task, $\mathcal{L}_i$:

$$\mathcal{L}_s^{TRL} = \sum_{i=1}^n \mathcal{L}_i(y_i^{TRL}, y_i) \qquad (5)$$

During test-time, the TRL output, $y_i^{TRL}$, is aligned with the main output, $y_i^{main}$, which lowers the upper bound of the objective eq. (3) on the target domain. Therefore, the total framework is trained with the Pseudo-label Prediction Loss $\mathcal{L}_t^{TRL}$ and the corresponding loss function is summarized as follows:

$$\mathcal{L}_t^{TRL} = \sum_{i=1}^n \mathcal{L}_i(y_i^{TRL}, y_i^{main}) \qquad (6)$$

In summary, during the train phase, the overall framework is trained by minimizing the following total loss (red arrows in the Fig. 2):

$$\mathcal{L}_s^{Total} = \mathcal{L}_s^{main} + \lambda^{TRL}\mathcal{L}_s^{TRL} + \lambda^{TP}\mathcal{L}_s^{TP}$$
$$= \sum_{i=1}^n \mathcal{L}_i(y_i^{main}, y_i) + \lambda^{TRL}\sum_{i=1}^n \mathcal{L}_i(y_i^{TRL}, y_i) + \lambda^{TP}\sum_{i=1}^n \mathcal{L}_i(y_i^{TP}, y_i) \qquad (7)$$

Each $\lambda^{TRL}$ and $\lambda^{TP}$ denotes the loss weight for $\mathcal{L}_s^{TRL}$ and $\mathcal{L}_s^{TP}$, respectively. In the test-time phase, the framework is only trained with the Pseudo-label Prediction Loss (blue arrows in the Fig. 2):

$$\mathcal{L}_t^{Total} = \mathcal{L}_t^{TRL} = \sum_{i=1}^n \mathcal{L}_i(y_i^{TRL}, y_i^{main}) \qquad (8)$$

## 4 EXPERIMENTS

In this section, we evaluate previous TTT methods against ours across multiple benchmarks. We also conduct an ablation study on each component of TR-TTT to analyze how the proposed methods effectively capture task relations and handle domain shifts during test-time training.

### 4.1 EXPERIMENTAL SETTINGS

**Datasets.** To evaluate the ability to reduce the domain gap on downstream tasks, which include both classification and regression, we utilize several existing multi-task benchmarks. We incorporate NYUD-v2 (Silberman et al., 2012), PASCAL-Context (Mottaghi et al., 2014), and Taskonomy (Zamir et al., 2018) in our TTT evaluation protocols. These datasets contain 4, 5, and 26 vision tasks, respectively. Following the typical protocol for TTT experimental settings, we use the shared task set between each pair of benchmarks, such as depth estimation, semantic segmentation, surface normal prediction, and edge detection for NYUD-v2 and Taskonomy. Also, for PASCAL-Context and taskonomy datasets, we use semantic segmentation, surface normal prediction, and edge detection. We select commonly used semantic labels for each dataset pair to simulate TTT protocols. Further details are provided in Appendix B.

**Baselines and Evaluation Protocols.** We compare our methods with previous test-time adaptation approaches, including TENT (Wang et al., 2020) and TIPI (Nguyen et al., 2023), as well as test-time training methods such as TTT (Sun et al., 2020), TTT++ (Liu et al., 2021), TTTFlow (Osowiechi et al., 2023), ClusT3 (Hakim et al., 2023), ActMAD (Mirza et al., 2023), and NC-TTT (Osowiechi et al., 2024). To evaluate the adaptation, we cover the domain shifts as follows: 1) Taskonomy→NYUD-v2, 2) Taskonomy→PASCAL-Context in the main paper, and 3) NYUD-v2→Taskonomy, 4) PASCAL-Context→Taskonomy in Appendix C. Since the TR-TTT can access multiple tasks in the source domain to learn their relations during training, we also allow the aforementioned baselines to access multi-task labels during training for a fair comparison. To achieve this, we similarly incorporate multi-task decoders to facilitate learning during training. This setup maximizes the potential of TTT methods, as learning multiple tasks in the source domain during training enhances generalizability at test-time by learning shared representations across tasks. During test-time, we evaluate all tasks simultaneously. For TTA, we adapt the model that was trained on multiple tasks in the source domain. To evaluate overall performance improvements during test time, we propose a metric, $\triangle_{TTT}$ for assessing TTT, motivated by Maninis et al. (2019). This metric measures averaged per-task performance improvements when applying TTT methods and is defined as: $\triangle_{TTT} = \frac{1}{n} \sum_{i=1}^{n} (-1)^{l_i} \frac{M_{TTT,i} - M_{b,i}}{M_{b,i}}$. In this equation, $M_{TTT,i}$ indicates the performance of task $i$ when TTT is applied, while $M_{b,i}$ represents the performance of task $i$ without TTT. The value $l_i = 1$ if a lower measure $M_i$ indicates better performance for task $i$, and $l_i = 0$ otherwise.

**Implementation Details.** For our experiments, we use resnet50 as an encoder and simple task-specific decoders that combines multi-layer features with convolutional layers. We also use a single convolutional layer for task-specific projection and a lightweight vision transformer for TRL. The Task Relation Learner (TRL) increases the network size by approximately $24.1\%$ when applied to ResNet50. The models are trained for 40,000 iterations on the source domain with a batch size of 8, and then sequentially trained on the target domain. We utilize the loss scales and loss functions that are commonly employed in existing multi-task learning literature (Yang et al., 2024; Ye & Xu, 2022b;a; Vandenhende et al., 2020; Zhang et al., 2019). We employ the Adam optimizer with a learning rate of $2 \times 10^{-5}$ and a weight decay of $1 \times 10^{-6}$, using a polynomial learning rate schedule.

### 4.2 EXPERIMENTAL RESULTS

**Comparison with Previous methods.** We compare TR-TTT with previous state-of-the-art TTT methods. Taskonomy is used as the source domain, and results for NYUD-v2 and Pascal-Context as target domains are presented in Table 1 and Table 2, respectively. Since each method converges at different rates, we select the point at which each method achieves its best TTT performance, averaged across all tasks, measured by $\triangle_{TTT}$ for a fair comparison. TR-TTT outperforms all other methods in both settings. A key observation is that the effectiveness of previous methods depends on the type of main task. Methods like TENT and NC-TTT, which rely on class-level clustering to reduce the domain gap, exhibit limited performance on regression tasks across both datasets. Even

Table 1: Comparison of multi-task performance from Taskonomy to NYUD-v2 across four different tasks for TR-TTT, against previous TTA and TTT methods.

| | Semseg (mIoU ↑) | Depth (RMSE ↓) | Normal (mErr ↓) | Edge (RMSE ↓) | $\triangle_{TTT}$ ↑ (%) |
|---|---|---|---|---|---|
| Base | 29.31 ±0.063 | 1.179 ±0.008 | 61.32 ±0.820 | 0.1443 ±0.71e-4 | +0.00 |
| Test-time Adaptation | | | | | |
| TENT (Wang et al., 2020) | 40.42 ±1.09 | 1.056 ±0.017 | 56.09 ±3.21 | 0.1441 ±0.21e-4 | +14.26 ±0.02 |
| TIPI (Nguyen et al., 2023) | 48.12 ±1.781 | 1.029 ±0.0651 | 55.71 ±0.493 | 0.1440 ±6.438e-5 | +21.57 ±0.329 |
| Test-time Training | | | | | |
| TTT (Sun et al., 2020) | 41.31 ±0.446 | 1.061 ±0.001 | 47.54 ±0.431 | 0.1440 ±2.83e-5 | +18.43 ±0.214 |
| TTT++ (Liu et al., 2021) | 43.97 ±1.110 | 1.107 ±0.015 | 46.71 ±0.064 | 0.1440 ±2.62e-5 | +20.05 ±1.228 |
| TTTFlow (Osowiechi et al., 2023) | 52.75 ±0.075 | 1.075 ±0.001 | 46.02 ±0.125 | 0.1442 ±4.73e-5 | +28.47 ±0.096 |
| ClusT3 (Hakim et al., 2023) | 41.88 ±2.058 | 1.102 ±0.014 | 46.52 ±0.007 | 0.1440 ±7.28e-6 | +18.44 ±1.450 |
| ActMAD (Mirza et al., 2023) | 27.62 ±0.161 | 1.193 ±0.008 | 56.53 ±2.307 | 0.1444 ±3.04e-5 | +0.203 ±1.256 |
| NC-TTT (Osowiechi et al., 2024) | 48.17 ±6.976 | 1.086 ±0.052 | 48.32 ±1.228 | 0.1440 ±2.83e-5 | +23.40 ±5.347 |
| TR-TTT (ours) | 59.37 ±0.152 | 1.052 ±7.5e-3 | 45.33 ±0.072 | 0.1441 ±5.1e-5 | +34.94 ±0.008 |

Table 2: Comparison of multi-task performance from Taskonomy to PASCAL-Context across three different tasks for TR-TTT, against previous TTA and TTT methods.

| | Semseg (mIoU ↑) | Normal (mErr ↓) | Edge (RMSE ↓) | $\triangle_{TTT}$ ↑ (%) |
|---|---|---|---|---|
| Base | 27.08 ±0.014 | 63.46 ±0.954 | 0.1185 ±0.71e-4 | 0.00 |
| Test-time Adaptation | | | | |
| TENT Wang et al. (2020) | 40.65 ±0.134 | 58.76 ±2.05 | 0.1183 ±0.09e-4 | +19.26 ±0.004 |
| TIPI Nguyen et al. (2023) | 43.01 ±0.0013 | 39.03 ±2.27e-4 | 0.1186 ±1.37e-8 | +32.41 ±0.002 |
| Test-time Training | | | | |
| TTT Sun et al. (2020) | 39.29 ±0.228 | 33.76 ±0.904 | 0.1183 ±0.04e-4 | +30.70 ±0.193 |
| TTT++ Liu et al. (2021) | 37.26 ±0.050 | 36.87 ±0.045 | 0.1183 ±0.24e-4 | +26.55 ±0.045 |
| TTTFlow Osowiechi et al. (2023) | 38.73 ±0.245 | 43.30 ±0.755 | 0.1184 ±4.72e-5 | +27.97 ±0.684 |
| ClusT3 Hakim et al. (2023) | 33.26 ±0.001 | 35.31 ±3.70e-5 | 0.1184 ±0.36e-8 | +22.43 ±0.001 |
| ActMAD Mirza et al. (2023) | 22.09 ±0.001 | 54.73 ±0.001 | 0.1188 ±0.19e-8 | -1.630 ±0.001 |
| NC-TTT Osowiechi et al. (2024) | 42.81 ±0.107 | 40.65 ±0.110 | 0.1184 ±0.77e-6 | +31.37 ±0.075 |
| TR-TTT (ours) | 45.42 ±0.19 | 41.41 ±0.63 | 0.1183 ±5.0e-5 | +34.20 ±0.11 |

in classification tasks such as semantic segmentation, feature-level adaptation methods that use class cluster information, like ClusT3, ActMad, and NC-TTT, show limited effectiveness. This suggests that these methods are better suited for simple classification tasks and struggle to generalize to more complex dense prediction tasks. The results also indicate that TTT performance heavily relies on the choice of unsupervised tasks selected for auxiliary training. In contrast, our TR-TTT method captures task relations and effectively incorporates them into the adaptation process, achieving superior performance across multiple tasks.

**Performance Over Time with Adaptation Iterations.** We evaluate the performance of each TTT method in an online manner over time with adaptation iterations. As shown in Fig. 3, the performance of TR-TTT continuously improves with an increasing number of time steps. In contrast, most other adaptation methods experience performance degradation during longer adaptation processes. This phenomenon has been frequently reported in previous research, such as TENT, which indicated that adaptation loss has a detrimental influence on learning the target task over longer adaptation periods. This is a crucial point in practice since we often do not know how many adaptation steps are needed during test-time. TR-TTT is less affected by this issue because it directly leverages the relations between the main tasks that the network is trying to adapt. Additionally, in situations where multiple tasks need to be adapted across domains, TR-TTT offers more advantages as it avoids problems related to differing convergence rates between tasks.

## 4.3 ABLATION STUDY

In this section, we present additional ablation experiments to evaluate each component of TR-TTT and their respective strategies. We assess the influence of the following components: (1) the Task Relation Learner (TRL), including the joint task prediction loss $\mathcal{L}^{TRL}$, (2) task-specific projection, including $\mathcal{L}^{TP}$, (3) feature masking applied to each task-specific latent vector, and (4) a comparison of results when using image reconstruction as an auxiliary task instead of the main tasks we aim

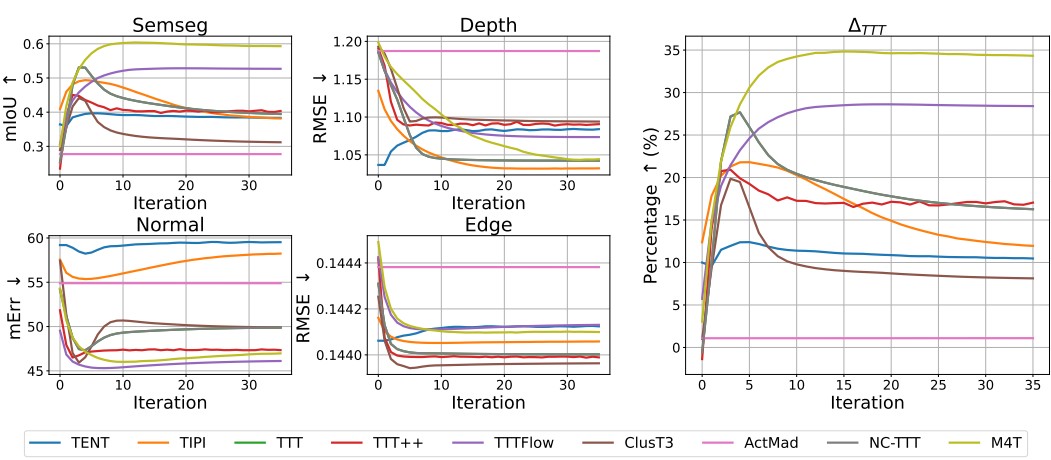

Figure 3: Comparison of previous TTA and TTT methods with our TR-TTT across time steps during test-time training. We evaluate the performance of each task and the overall TTT performance, denoted as $\triangle_{TTT}$, under the domain shift from Taskonomy to NYUD-v2.

Table 3: Ablation study for individual components of TR-TTT.

| Benchmark | Taskonomy → NYUD-v2 | | | | | Taskonomy → PASCAL-Context | | | | |
|---|---|---|---|---|---|---|---|---|---|---|
| TRL | - | ✓ | ✓ | ✓ | ✓ | - | ✓ | ✓ | ✓ | ✓ |
| Task-specific Projection | - | - | ✓ | - | ✓ | - | - | ✓ | - | ✓ |
| Feat. Masking | - | - | - | ✓ | ✓ | - | - | - | ✓ | ✓ |
| Image Recon. Task | - | - | - | ✓ | - | - | - | - | ✓ | - |
| $\triangle_{TTT} \uparrow (\%)$ | +0.00 | +26.16 | +33.79 | +26.76 | +34.94 | +0.00 | +28.32 | +32.27 | +26.43 | +34.20 |

to adapt for the TTT branch. In Table 3, we show the performance improvements in TTT based on different combinations of these components, with improvements measured relative to results without any TTT methods, denoted as $\triangle_{TTT} \uparrow (\%)$.

**Ablation on Each Component.** In the TR-TTT framework, using the TRL alone effectively reduces the domain gap, leading to performance improvements of 26.16% for NYUD-v2 and 28.32% for PASCAL-Context. In this scenario, the TRL captures task relations by utilizing shared representations across multiple tasks, rather than task-specific latent vectors. When task-specific projection is introduced to extract task-specific latent vectors for the TRL, the performance further improves, suggesting that task relations are more effectively captured with distinct task-specific information. Additionally, the feature masking strategy, which is analyzed in the following subsection, provides further performance gains, although most of the improvements are driven by the TRL and task-specific projection. Lastly, to validate our TTT strategy, we focused on capturing the task relations we want to adapt. Instead of using an entirely different auxiliary task like image reconstruction, we integrated image reconstruction into our TTT branch. In this setup, the TRL predicts the reconstructed image instead of task labels. The learned information from image reconstruction resulted in significantly poorer TTT performance compared to our approach. This highlights that using auxiliary tasks, such as image reconstruction, does not necessarily ensure the inclusion of useful information for downstream tasks, especially in the context of domain shift.

**Masking Strategy and Masking Ratio.** To evaluate which masking strategy $\mathcal{M}$ for task-specific latent vectors $\tilde{z}_i = \mathcal{M}_i(z_i)$ would be beneficial for learning inter-task relations, we select several candidates for masking strategies to assess their influence, as shown in Fig. 4. We consider four scenarios: (a) we randomly mask each task-specific latent vector $z_i$, (b) we mask them without overlap across tasks, (c) we mask identical patches, which are randomly chosen for all tasks, and (d) we randomly select task sets and entirely mask their task-specific latent vectors. As shown in Table 4, (c) Same for All shows the best performance, thus we adopt this strategy for our methods. We guess there are two reasons why (c) produces the best performance compared to the other strategies. First, although the task-specific latent vector $z_i$ is derived from a task-specific projection, it may still contain shared representations from other tasks. In such cases, using each task's latent

Table 4: Ablation study on the impact of different masking strategies, as described in Fig. 4.

| Method | (a) Random | (b) Not Overlap | (c) Same for All | (d) Hide specific Tasks |
|---|---|---|---|---|
| $\triangle_{TTT} \uparrow (\%)$ | +32.76 | +32.56 | +34.94 | +33.06 |

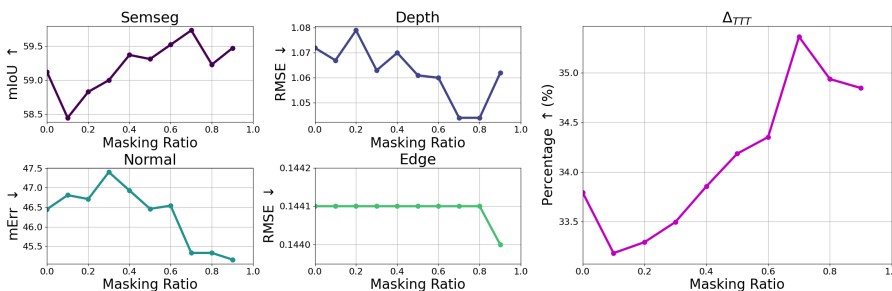

(a) Random     (b) Not Overlap     (c) Same for all     (d) Hide specific Tasks

Figure 4: Candidates for the masking strategy $\mathcal{M}_i$ applied to task-specific latent vectors, denoted as $\tilde{z}_i = \mathcal{M}_i(z_i)$. Each task-specific latent vector $z_i$ is represented by separate large squares, with the unmasked portions shaded in black. In (a), we randomly select patches for masking. In (b), we mask without overlap between tasks, represented as $\mathcal{M}_1 \cap \mathcal{M}_2 \cap \cdots \cap \mathcal{M}_n = \emptyset$. In (c), we apply the same masking strategy across all task-specific latent spaces, denoted as $\mathcal{M}_1 = \mathcal{M}_2 = \cdots = \mathcal{M}_n$. In (d), we completely mask the latent vector of a specific task, indicated as $\mathcal{M}_i = \emptyset$ for some $i$.

Figure 5: Ablation study on the masking ratio of TR-TTT. We evaluate the performance under the domain shift from Taskonomy to NYUD-v2.

vector for prediction results in trivial predictions by the TRL. Second, predicting the task label for a masked patch from the unmasked patch encourages the TRL to capture spatially global information across different task-specific latent vectors. If the TRL has access to the same patch location from another task's latent vector, it might merely memorize the style transfer between these vectors, which would negatively affect generalization. In Fig. 5, we evaluate the influence of the masking ratio for the adopted masking strategy (c) on the performance of tasks during test-time. The overall TTT performance improves as the masking ratio increases, peaking at approximately 0.7 to 0.8. It is noteworthy that the overall trend is quite consistent across tasks.

## 5 CONCLUSION

In this paper, we introduce Task Relation Learning for Test-time Training (TR-TTT) to address the distribution gap between source and target domains during adaptation. We demonstrate that understanding task relations is crucial for successful adaptation in TTT. By employing a Task Relation Learner to capture these relations as conditional probabilities, our approach enables the network to predict the labels of target tasks using information from other task-specific latent spaces. This innovative strategy allows TR-TTT to manage distribution shifts more effectively and enhances post-adaptation performance across a range of tasks, including both classification and regression. We validated our approach through extensive experiments using conventional multi-task benchmarks integrated with established TTT protocols. The empirical results indicate a significant performance improvement compared to state-of-the-art methods, confirming the effectiveness of our framework.

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

# A DERIVATIONS OF PROPOSITION 1

For simplicity, denote the task-specific latent space as $\{z_{t,i}\}_{i=1}^n$ and its masked version as $\{\tilde{z}_{t,i}\}_{i=1}^n$.

$$d(\theta, p(\{z_{t,i}\}_{i=1}^n, y_j)) - d(\theta, p(\{\tilde{z}_{t,i}\}_{i=1}^n, y_j)) \tag{9}$$

$$= \mathbb{E}_{p(\{z_{t,i}\}_{i=1}^n)}[d[p(y_j|\{z_{t,i}\}_{i=1}^n), p_\theta(y_j|\{z_{t,i}\}_{i=1}^n)]] \tag{10}$$

$$- \mathbb{E}_{p(\{\tilde{z}_{t,i}\}_{i=1}^n)}[d[p(y_j|\{z_{t,i}\}_{i=1}^n), p_\theta(y_j|\{\tilde{z}_{t,i}\}_{i=1}^n)]] \tag{11}$$

$$= \mathbb{E}_{p(\{z_{t,i}\}_{i=1}^n, \{\tilde{z}_{t,i}\}_{i=1}^n)}[d[p(y_j|\{z_{t,i}\}_{i=1}^n), p_\theta(y_j|\{z_{t,i}\}_{i=1}^n)]] \tag{12}$$

$$- \mathbb{E}_{p(\{z_{t,i}\}_{i=1}^n, \{z_{t,i}\}_{i=1}^n \{\tilde{z}_{t,i}\}_{i=1}^n)}[d[p(y_j|\{z_{t,i}\}_{i=1}^n), p_\theta(y_j|\{\tilde{z}_{t,i}\}_{i=1}^n)]] \tag{13}$$

$$\leq \mathbb{E}_{p(\{z_{t,i}\}_{i=1}^n, \{\tilde{z}_{t,i}\}_{i=1}^n)}[d[p_\theta(y_j|\{z_{t,i}\}_{i=1}^n), p_\theta(y_j|\{\tilde{z}_{t,i}\}_{i=1}^n)]] \tag{14}$$

$$+ \mathbb{E}_{p(\{z_{t,i}\}_{i=1}^n, \{\tilde{z}_{t,i}\}_{i=1}^n)}[d[p(y_j|\{z_{t,i}\}_{i=1}^n), p(y_j|\{\tilde{z}_{t,i}\}_{i=1}^n)]] \tag{15}$$

The eq. (15) follows from the triangle inequality.

Rearranging the above equation results in the following inequality.:

$$d(\theta, p(\{z_{t,i}\}_{i=1}^n, y_j)) \leq d(\theta, p(\{\tilde{z}_{t,i}\}_{i=1}^n, y_j)) \tag{16}$$

$$+ \mathbb{E}_{p(\{z_{t,i}\}_{i=1}^n, \{\tilde{z}_{t,i}\}_{i=1}^n)}[d[p_\theta(y_j|\{z_{t,i}\}_{i=1}^n), p_\theta(y_j|\{\tilde{z}_{t,i}\}_{i=1}^n)]] \tag{17}$$

$$+ \mathbb{E}_{p(\{z_{t,i}\}_{i=1}^n, \{\tilde{z}_{t,i}\}_{i=1}^n)}[d[p(y_j|\{z_{t,i}\}_{i=1}^n), p(y_j|\{\tilde{z}_{t,i}\}_{i=1}^n)]] \tag{18}$$

In the multi-task setting, we apply eq. (18) to each task as follows:

$$\sum_{j=1}^n d(\theta, p(\{z_{t,i}\}_{i=1}^n, y_j)) \leq \sum_{j=1}^n d(\theta, p(\{\tilde{z}_{t,i}\}_{i=1}^n, y_j)) \tag{19}$$

$$+ \sum_{j=1}^n \mathbb{E}_{p(\{z_{t,i}, \tilde{z}_{t,i}\}_{i=1}^n)}[d[p_\theta(y_j|\{z_{t,i}\}_{i=1}^n), p_\theta(y_j|\{\tilde{z}_{t,i}\}_{i=1}^n)]] \tag{20}$$

$$+ \sum_{j=1}^n \mathbb{E}_{p(\{z_{t,i}, \tilde{z}_{t,i}\}_{i=1}^n)}[d[p(y_j|\{z_{t,i}\}_{i=1}^n), p(y_j|\{\tilde{z}_{t,i}\}_{i=1}^n)]] \tag{21}$$

$$\leq \sum_{j=1}^n d(\theta, p(\{\tilde{z}_{t,i}\}_{i=1}^n, y_j)) \tag{22}$$

$$+ \sum_{j=1}^n \mathbb{E}_{p(\{z_{t,i}, \tilde{z}_{t,i}\}_{i=1}^n)}[d[p_\theta(y_j|\{z_{t,i}\}_{i=1}^n), p_\theta(y_j|\{\tilde{z}_{t,i}\}_{i=1}^n)]] \tag{23}$$

$$+ \sum_{j=1}^n \mathbb{E}_{p(\{z_{s,i}, \tilde{z}_{s,i}\}_{i=1}^n)}[d[p(y_j|\{z_{s,i}\}_{i=1}^n), p(y_j|\{\tilde{z}_{s,i}\}_{i=1}^n)]] \tag{24}$$

The inequality between eq. (21) and eq. (24) holds due to assumption 1, which states that task relations are preserved between tasks and their masked versions. With an adequate masking ratio, the joint MAE sufficiently captures the task relations in the source domain, and eq. (24) approaches zero, as this is the objective during training in the source domain.

Therefore, the following inequality holds:

$$\sum_{j=1}^n d(\theta, p(\{z_{t,i}\}_{i=1}^n, y_j)) \leq \sum_{j=1}^n d(\theta, p(\{\tilde{z}_{t,i}\}_{i=1}^n, y_j)) \tag{25}$$

$$+ \sum_{j=1}^n \mathbb{E}_{p(\{z_{t,i}, \tilde{z}_{t,i}\}_{i=1}^n)}[d[p_\theta(y_j|\{z_{t,i}\}_{i=1}^n), p_\theta(y_j|\{\tilde{z}_{t,i}\}_{i=1}^n)]] \tag{26}$$

## B  ADDITIONAL EXPERIMENTAL DETAILS

**Experimental Settings.** In the training phase within the source domain, we utilize the Adam optimizer (Kingma & Ba, 2014) with a polynomial decay for the learning rate. We set the learning rate to $2 \times 10^{-5}$ and the weight decay to $1 \times 10^{-6}$ for training the networks. The batch size is 8, and we perform 60,000 iterations for training. During test time, we adopt the SGD optimizer to ensure stable convergence with the TTT loss. The learning rate remains the same, but we reduce the loss scale for TTT to approximately 0.01. During test time, we update the network for each batch of data for up to 40 steps in an online manner.

Table 5: Hyperparameters for experiments.

| Hyperparameter | Value |
|---|---|
| ⌊ Scheduler | Polynomial Decay |
| ⌊ Minibatch size | 8 |
| ⌊ Backbone | ResNet50 (He et al., 2016) |
| ⌊ Learning rate | 0.00002 |
| ⌊ Weight Decay | 0.000001 |
| Train Time Training | |
| ⌊ Optimizer | Adam (Kingma & Ba, 2014) |
| ⌊ Number of iterations | 60000 |
| ⌊ Learning rate | 0.00002 |
| ⌊ Weight Decay | 0.000001 |
| Test Time Training | |
| ⌊ Optimizer | SGD |
| ⌊ Minibatch size | 8 |
| ⌊ Number of steps | 40 |

**Metrics.** For semantic segmentation, we utilize the mean Intersection over Union (mIoU) metric. The performance of surface normal prediction was measured by calculating the mean angle distances between the predicted output and the ground truth. To evaluate depth estimation and edge detection, we use the Root Mean Squared Error (RMSE).

**Datasets.** To implement TTT in semantic segmentation tasks on different datasets (Taskonomy ↔ NYUD-v2, Taskonomy ↔ PASCAL-Context), we find shared class labels in each of the two datasets. For Taskonomy ↔ NYUD-v2, we use 6 shared classes: `table`, `tv`, `toilet`, `sofa`, `potted plant`, `chair`. For Taskonomy ↔ PASCAL-Context, we use 7 class labels: `refridgeator`, `table`, `toilet`, `sofa`, `bed`, `sink`, `chair`. We use the split of train/test following the common multi-task benchmarks, NYUD-v2, PASCAL-Context and Taskonomy. In the case of NYUD-v2, we utilize 795 images for training and reserve 654 images for test-time training. With PASCAL-Context, 4,998 images are employed during training, and 5,105 images are used for test-time training. For Taskonomy, we leverage 295,521 images for training and apply 5,451 images during test-time.

## C  ADDITIONAL EXPERIMENTS

**Comparison with Previous Methods in Different Scenarios.** We compare TR-TTT with previous state-of-the-art TTT methods in different scenarios, using NYUD-v2 and PASCAL-Context as the source domains and Taskonomy as the target domain. The results are presented in Tables 6 and 7, respectively. For a fair comparison, we select the point at which each method achieves its best TTT performance, averaged across all tasks, as measured by $\triangle_{TTT}$. Since NYUD-v2 and PASCAL-Context have smaller datasets, the overall TTT performance is lower compared to scenarios where Taskonomy is used as the source domain. The proposed TR-TTT still demonstrates comparable performance in these scenarios.

Table 6: Comparison of multi-task performance from NYUD-v2 to Taskonomy across four different tasks for TR-TTT, against previous TTA and TTT methods.

| | Semseg (mIoU ↑) | Depth (RMSE ↓) | Normal (mErr ↓) | Edge (RMSE ↓) | $\triangle_{TTT}$ ↑ (%) |
|---|---|---|---|---|---|
| Base | 48.21 ±1.58 | 0.0507 ±0.0002 | 27.60 ±0.12 | 0.3058 ±2.40e-4 | 0.00 |
| *Test Time Adaptation* | | | | | |
| TENT (Wang et al., 2020) | 39.75 ±1.20 | 0.0634 ±0.00 | 37.49 ±0.35 | 0.3084 ±4.94e-4 | -19.76 ±0.23 |
| TIPI (Nguyen et al., 2023) | 47.03 ±7.08e-4 | 0.0514 ±3.55e-8 | 28.40 ±1.49e-5 | 0.3052 ±3.32e-8 | -1.639 ±0.0003 |
| *Test Time Training* | | | | | |
| TTT (Sun et al., 2020) | 49.66 ±0.412 | 0.0523 ±4.12e-3 | 31.96 ±0.119 | 0.3094 ±6.27e-4 | -4.276 ±0.5720 |
| TTT++ (Liu et al., 2021) | 39.29 ±1.089 | 0.0595 ±1.29e-3 | 36.53 ±0.416 | 0.3132 ±2.61e-3 | -17.65 ±0.6608 |
| TTTFlow (Osowiechi et al., 2023) | 48.56 ±0.300 | 0.054 ±1.297e-4 | 34.36 ±0.1425 | 0.3086 ±9.01e-5 | -7.73.00 ±0.356 |
| ClusT3 (Hakim et al., 2023) | 51.14 ±1.757 | 0.0516 ±3.75e-4 | 30.03 ±0.431 | 0.3065 ±4.60e-4 | -1.164 ±1.449 |
| ActMAD (Mirza et al., 2023) | 55.04 ±0.73e-4 | 0.0506 ±0.58e-8 | 27.88 ±0.07e-4 | 0.3081 ±1.10e-9 | +3.173 ±4.1326e-5 |
| NC-TTT (Osowiechi et al., 2024) | 49.95 ±0.653 | 0.0516 ±0.046e-5 | 29.95 ±0.042 | 0.3093 ±1.01e-4 | -1.957 ±0.3619 |
| TR-TTT (ours) | 53.12 ±0.134 | 0.0511 ±1.93e-4 | 27.58 ±0.1044 | 0.3089 ±3.54e-5 | +2.13 ±0.071 |

Table 7: Comparison of multi-task performance from PASCAL-Context to Taskonomy across three different tasks for TR-TTT, against previous TTA and TTT methods.

| | Semseg (mIoU ↑) | Normal (mErr ↓) | Edge (RMSE ↓) | $\triangle_{TTT}$ ↑ (%) |
|---|---|---|---|---|
| Base | 50.94 ±0.663 | 31.27 ±0.071 | 0.3032 ±0.141e-4 | 0.00 |
| *Test Time Adaptation* | | | | |
| TENT Wang et al. (2020) | 44.68 ±0.353 | 42.32 ±0.183 | 0.3269 ±0.21e-4 | -0.184 ±0.007 |
| TIPI Nguyen et al. (2023) | 51.48 ±0.0012 | 32.33 ±4.18e-5 | 0.3031 ±1.49e-7 | -0.766 ±7.74e-4 |
| *Test Time Training* | | | | |
| TTT Sun et al. (2020) | 48.00 ±2.661 | 37.77 ±2.214 | 0.3048 ±3.60e-4 | -9.002 ±4.141 |
| TTT++ Liu et al. (2021) | 38.66 ±0.309 | 39.81 ±0.385 | 0.3050 ±6.48e-4 | -17.33 ±0.1945 |
| TTTFlow Mirza et al. (2023) | 51.55 ±0.395 | 34.60 ±0.120 | 0.3042 ±1.45e-3 | -3.258 ±0.0289 |
| ClusT3 Osowiechi et al. (2023) | 49.67 ±0.648 | 35.22 ±0.157 | 0.3019 ±1.90e-4 | -4.904 ±0.2365 |
| ActMad Hakim et al. (2023) | 51.79 ±0.803 | 31.10 ±0.124 | 0.3031 ±1.02e-4 | +0.744 ±0.3821 |
| NC-TTT Osowiechi et al. (2024) | 48.78 ±0.510 | 32.86 ±2.220 | 0.3040 ±1.28e-3 | -3.187 ±2.841 |
| TR-TTT (ours) | 53.18 ±0.315 | 31.50 ±0.0789 | 0.3036 ±0.0002 | +1.179 ±0.175 |

Table 8: We compare the TTT performance of Taskonomy as the source domain and NYUD-v2 as the target domain across four tasks for TR-TTT, analyzing both single-task and multi-task scenarios.

| | Semseg (mIoU ↑) | Depth (RMSE ↓) | Normal (mErr ↓) | Edge (RMSE ↓) | $\triangle_{TTT}$ ↑ (%) |
|---|---|---|---|---|---|
| Base | 29.31 ±0.063 | 1.179 ±0.008 | 61.32 ±0.820 | 0.1443 ±0.71e-4 | +0.00 |
| TR-TTT (single) | 59.37 ±0.152 | 1.052 ±7.5e-3 | 45.33 ±0.072 | 0.1441 ±5.1e-5 | - |
| TR-TTT | 59.37 ±0.152 | 1.052 ±7.5e-3 | 45.33 ±0.072 | 0.1441 ±5.1e-5 | +34.94 ±0.008 |

Table 9: We compare the TTT performance of Taskonomy as the source domain and PASCAL-Context as the target domain across three tasks for TR-TTT, analyzing both single-task and multi-task scenarios.

| | Semseg (mIoU ↑) | Normal (mErr ↓) | Edge (RMSE ↓) | $\triangle_{TTT}$ ↑ (%) |
|---|---|---|---|---|
| Base | 27.08 ±0.014 | 63.46 ±0.954 | 0.1185 ±0.71e-4 | 0.00 |
| TR-TTT (single) | 43.28 ±0.37 | 46.91 ±0.075 | 0.1185 ±9.6e-6 | - |
| TR-TTT | 45.42 ±0.19 | 41.41 ±0.63 | 0.1183 ±5.0e-5 | +34.20 ±0.11 |

**Evaluation of TR-TTT Using Single Task for Adaptation.** Evaluating learned task relations is a crucial aspect of our framework. We assess performance improvements during adaptation by allowing access to each single-task label in the source domain. In this scenario, TLR predicts the single-task label using a single latent vector corresponding to that task. As shown in the results (see Tables 8 and 9), the single-task scenario exhibits significantly lower TTT performance, highlighting the importance of task relations for TTT.

