# OpenReview forum: "Learning Task Relations for Test-Time Training"
_ICLR.cc/2025/Conference — ICLR 2025 Conference Withdrawn Submission_

### Official Review · Reviewer_BfXM · 2024-10-31

**Soundness:** 2
**Presentation:** 3
**Contribution:** 2
**Rating:** 5
**Confidence:** 4

**Summary:**

This paper proposes TR-TTT, a test-time training approach that leverages task relations to address domain shifts. By capturing task relations as conditional probabilities and using a Task Relation Learner (TRL), TR-TTT enables the network to predict target task labels using information from other task-specific latent spaces. This strategy effectively manages distribution shifts and improves post-adaptation performance across various tasks, including classification and regression.

**Strengths:**

1. TR-TTT represents a significant departure from existing TTT methods that rely on auxiliary tasks. By focusing on task relations, it provides a more robust and generalizable solution to domain shifts.

2. The paper conducts extensive experiments on multiple multi-task benchmarks, including both classification and regression tasks. The evaluation protocols are well-designed, and the results demonstrate the superiority of TR-TTT over existing methods.

3. The paper is well-organized and follows a logical flow. Each section is clearly defined and builds upon the previous sections.

**Weaknesses:**

1.While the paper presents Task Relation Learning (TRL) as a novel approach, it builds heavily on existing multi-task learning (MTL) techniques. The use of task-specific latent spaces and conditional probabilities for inter-task relationships is not entirely new.

2.This paper was validated on a multi task benchmark when exploring task relation. I think a more interesting setting is that, for example, all methods are compared only on the classification task, while the author uses different auxiliary tasks such as rotation angle prediction, contrastive learning, and so on together. I want to know if task relation can improve the performance of the model under this setting.

**Questions:**

1. Can you elaborate on the specific differences between TRL and existing MTL methods? How does TRL uniquely leverage task relations for test-time training?
2. Can you provide a more detailed theoretical analysis of the assumptions made in the paper, particularly the preservation of task relations across domains? What conditions must be met for these assumptions to hold?
3. I would like to know if other baselines also use the same data as TRL during training, that is, multi task data for joint training; If so, I would like to know how effective other baselines would be in training using only the corresponding task data on different tasks.

**Details Of Ethics Concerns:**

No.

---

### Official Review · Reviewer_eL1j · 2024-10-31

**Soundness:** 1
**Presentation:** 1
**Contribution:** 1
**Rating:** 3
**Confidence:** 4

**Summary:**

The paper presents a new approach for test-time training (TTT) so as to deal with distribution shift during test time. The idea, if I understand correctly, is to exploit some auxiliary tasks in the source domain and combine the task-specific features for prediction. Empirical successes are presented.

**Strengths:**

The idea of exploiting multi-task learning to tackle distribution shift and exploiting it in the latent space is valid.

**Weaknesses:**

The paper is poorly presented, the notations and concepts are messy. For example, in 3.1, ${\mathcal Y}$ represents a random variable, then ${\mathcal Y}_s={\mathcal Y}_t$ is meant to say that the source domain and target domain have the same label space, but note that as random variables they are not the same.  (In 3.2, ${\mathcal Z}_i$  refers to both a random variable, and a subspace.)

Further,
${\mathcal Z}_{s, i}$ is used without definition. There are a number of cases like this,  in which I have to guess.

The model architecture is not clear. From Figure 1 and its description, I do not see how task-specific features impact the "main" task.

Trivial assumptions. Since the authors use the phrase "task-specific projections", I assume that for each task i, there is task-specific mapping $f_i$ such that $\hat{\mathcal Z}_{s, i}$ (when interpreted a random variable) is $f_i({\mathcal X}_s)$. If this interpretation is correct, one may trivially choose each $f_i$ to be a constant function to make Assumption 1 holds (in which case the mutual information on both sides are zero).  — That is, additional assumptions of $f_i$’s are needed in order to make this Assumption 1 useful.

Proposition 1 is poorly stated. What does it mean by a distance/metric between $\theta$ and a distribution? If I interpret $\theta$ as a distribution, it should be a conditional distribution, but the other distribution is not in the conditional form. It is not even clear what the proposition is intended to mean, and hence I can not decide if it is correct, at all.

The notion of TTT or test-time adaptation really falls into the setting of unsupervised domain adaptation (UDA), if we treat the unlabelled test data as unlabelled training data in the target domain. The only thing extra is the availability of multiple auxiliary tasks. But the vast literature of UDA is completely ignored from citation and discussion. Existing UDA algorithms can be easily adapted to the multi-task setting of this paper , but none of those algorithms are compared.

The paper is largely motivated by some (possibly) valid intuition, but its depth is shallow and its insight is thin, in addition to various errors and lack of clarity.

**Questions:**

See weakness

---

### Official Review · Reviewer_a3s4 · 2024-11-03

**Soundness:** 3
**Presentation:** 4
**Contribution:** 3
**Rating:** 6
**Confidence:** 4

**Summary:**

The paper introduces a Task Relation Learner (TRL) to model task relations as conditional probabilities, enabling more effective multi-task adaptation. The proposed method shows superior performance over existing methods across both classification and regression tasks.

**Strengths:**

1.	The motivation is well-founded.
2.	The writing is clear, with the assumptions, proposed algorithm, and illustrations all presented effectively.

**Weaknesses:**

The novelty appears limited, as this work primarily combines an MTL approach within a TTT scenario.

**Questions:**

Possible typos in the equations on lines 208 and 237.

---

### Official Review · Reviewer_84Xu · 2024-11-04

**Soundness:** 3
**Presentation:** 3
**Contribution:** 2
**Rating:** 6
**Confidence:** 3

**Summary:**

This paper proposes the framework Task Relation Learning for Test-time Training (TR-TTT) that helps learn multiple tasks in test-time training, usually auxiliary self-supervised tasks and target tasks.
task relations as conditional probabilities by predicting
The task relations are modeled as conditional probabilities, predicting the label of a target task based on the latent spaces of task-specific features.
They demonstrate improved performance for TR-TTT on some of the multitask benchmarks for regression and classification tasks.

**Strengths:**

* TR-TTT helps in reducing domain gaps across tasks
* Projecting the latent z into the task-specific latent z_i, and the idea of masking task-specific feature is interesting

**Weaknesses:**

* Lacking motivation as to why task relation learning is relevant for TTT and not multi-task learning in general
* The proposed approach does not perform better across most of the compared settings

**Questions:**

* Ablation numbers show that using all the components is not always helpful. Do the authors have any reasoning behind why some modules do worse for some settings?
* Why is the proposed task relation learning relevant for TTT and not multi-task learning in general? Is there any non-triviality in applying it for TTT? If so, can the authors elaborate on it?
* Are the lambda^TRL and lambda^TP tuned using validation set?

**Minor comments**
* The highest numbers in the experiments need to be bold or highlighted.

---

### Official Review · Reviewer_XD3e · 2024-11-04

**Soundness:** 3
**Presentation:** 3
**Contribution:** 2
**Rating:** 5
**Confidence:** 3

**Summary:**

The paper studies the problem of test-time training, which aims to update a model's weight at test-time to better adapt to the target domain of interest. The paper proposes a framework, TR-TTT, wherein the core idea is to learn task relations at training time, and in turn utilize the task relations, which is assumed to generalize across domains, at test-time to further adapt the model on the target domains. Empirical results show improvements over existing test-time training approaches.

**Strengths:**

- The paper presents an interesting idea that uses task relation prediction as a generalizable auxiliary objective for test-time training.
- The paper is generally clear.
- Empirical results are promising compared to existing methods. The ablation studies also give some interesting insights on the design of different components of TR-TTT, and how to select the masking strategies.

**Weaknesses:**

- The problem setting assumes the underlying availability of multi-tasks, which might not be available for all application scenarios. In addition, the authors do not seem to mention how to select the set of tasks to start with. How robust is the method to the selection of the set of tasks? It would be nice to see an ablation on the effect of task selection.
- The current baselines considered in the paper focus mostly on test-time training/adaptation methods. However, large-scale pretrained models have also shown to be strong in adapting to various target domains. As large-scale pretrained models are very common today, it would be great to discuss/compare the proposed approach to the pretrain-then-finetune (or even pretrain-then-zeroshot) paradigms. Specifically, multi-task training has also powered large-scale pretraining such as [1, 2]. In [1], they also consider the use of MAE in pretraining. How does TR-TTT compare to these works?
- It is not very clear to me why using only pseudo-label at test time works in Eq 8. Can the authors provide some intuition (or even better, theoretical justification) for this? In addition, if we assume TRL has learned the task-relation during training, why do we not freeze TRL's parameters during test-time? Have the authors done some experiments on these?

[1] 4M: Massively Multimodal Masked Modeling. Mizrahi et al. 2023.

[2] CLIP meets Model Zoo Experts: Pseudo-Supervision for Visual Enhancement. Salehi et al. 2023

**Questions:**

See above.

**Details Of Ethics Concerns:**

No immediate ethics concerns.

---

### Note · Authors · 2024-11-15

**Comment:**

Thank you to all reviewers for their sincere feedback and efforts. Unfortunately, we have decided to withdraw our paper.

**Withdrawal Confirmation:**

I have read and agree with the venue's withdrawal policy on behalf of myself and my co-authors.